# An Online Cross-Sectional Survey of Complementary Feeding Practices during the COVID-19 Restrictions in Poland

**DOI:** 10.3390/nu13093196

**Published:** 2021-09-14

**Authors:** Andrea Horvath, Agata Stróżyk, Piotr Dziechciarz, Hania Szajewska

**Affiliations:** Department of Paediatrics, Medical University of Warsaw, 02-091 Warsaw, Poland; ahorvath@wum.edu.pl (A.H.); piotrdz@hotmail.com (P.D.); hszajewska@wum.edu.pl (H.S.)

**Keywords:** infant, child, COVID-19, diet, breastfeeding, feeding behavior

## Abstract

This cross-sectional online survey performed in Poland aimed to improve understanding of how COVID-19 pandemic restrictions affected complementary feeding practices among parents of infants aged 4 to 12 months. Self-selected parents were recruited through the internet. The anonymous questionnaire was opened during two intervals during COVID-19 restrictions. The primary outcome was an assessment of sources of information and infant feeding practices in the context of COVID-19 restrictions. Data from 6934 responders (92.2% mothers) were analyzed. Most responders received information from multiple sources, with other parents, family members, or friends being the most frequently reported (48.6%), followed by webinars and experts’ recommendations (40.8%). COVID-19 restrictions largely did not impact the method of feeding, changes in feeding patterns, or complementary feeding introduction, although the latter was more likely to be impacted in families with average versus the best financial situations. Multivariate logistic regression analysis also most consistently showed that parents with a tertiary education and living in a city above 500 k were at higher odds of using webinars/experts’ recommendations, internet/apps, and professional expert guides and lower odds of claiming no need to deepen knowledge. This study clarifies major issues associated with complementary feeding practices during the implementation of COVID-19 restrictions in Poland.

## 1. Introduction

On 11 March 2020, the World Health Organization (WHO) declared the COVID-19 pandemic [1]. Following the WHO announcement, a range of measures to slow the spread of the virus (‘flatten the curve’) was imposed at national and international levels. However, these generally included combinations of stay-at-home restrictions, travel bans, school closures, closures of places of entertainment, and restrictions on public and private gatherings. In Poland, the first case of COVID-19 was reported on 4 March 2020 [2]. Soon after, the Polish government imposed the first strict so-called lockdown to limit the spread of the virus. The COVID-19 restrictions included covering the mouth and nose (i.e., by mask), keeping a safe distance between people of at least 2 m, and proper room ventilation. Restrictions also included limited or no access to many facilities such as hairdressers, shopping centers, cinemas, museums, restaurants, swimming pools, and gyms. Childcare centers and schools were closed for an extended time. Depending on the situation, COVID-19 restrictions were gradually relaxed or tightened. However, even at the time of writing this manuscript (July–August 2021), some restrictions are still in place. 

The COVID-19 pandemic resulted in unprecedented challenges with respect to healthcare. At least in Poland, access to routine pediatric in-person health services was disrupted during the strict lockdown, and patients relied on telehealth (audio and video consultations), also called telemedicine. However, while acknowledging the benefits of telehealth, concerns were expressed that it may compromise patient safety, particularly in pediatrics. The lack of proper physical examination was also one of the concerns [3]. Consequently, in November 2020, more detailed recommendations regarding when telemedicine in pediatrics (also in other specialties) may and may not be practiced were formulated [4]. Finally, since March 2021, telehealth can no longer be provided to children younger than 6 years of age. One exception is a follow-up consultation that was proceeded by a physical examination of the patient, which can be provided without repeated physical examination [5].

As recently reviewed [6], reduced income, limited financial resources, limited access to fresh and safe foods, limited access to healthcare, and interrupted education, all of which may occur during COVID-19 restrictions, may have an impact on infant feeding practices. They may to result in poor dietary intake and, consequently, compromised maternal and child nutrition, including breastfeeding and complementary feeding practices. 

With regard to breastfeeding, one Italian prospective cohort study of 204 mother-baby dyads found a decrease in the rate of exclusively breastfed infants during lockdown compared to a retrospective population of 306 mother-baby dyads admitted in 2018 [7]. Other recent studies that assessed the impact of COVID-19 restrictions on breastfeeding were performed in the United Kingdom (UK). The first of these studies [8] involved an online survey including 1219 mothers of breastfed infants aged 0–12 months. This study found that 27.0% of mothers had an issue receiving support with regard to breastfeeding, with some stopping breastfeeding before they were ready. However, 41.8% of the women had a positive breastfeeding experience during the COVID-19 pandemic. In the group with a negative experience, there were more mothers with a lower level of education and of Black or other minority ethnic groups for whom the pandemic might be more challenging. In the second study, a preliminary analysis of the United Kingdom COVID-19 New Mum Study [9] showed that only 13% of mothers reported changes to infant feeding practices with regard to COVID-19 restrictions during the UK lockdown. Moreover, similarly as in the previous UK study, some women reported positive breastfeeding behaviors such as increases in the frequency and duration of feeds (30% and 17%, respectively). However, some of the women reported a negative impact of COVID-19 on breastfeeding experiences such as decreases in the frequency and duration of feeds (10% and 15%, respectively). Similar to the other UK study, the authors state that the discrepancy in breastfeeding experiences may be associated with the more profound impact of COVID-19 restrictions on Black, other ethnic minorities, and other disadvantaged groups. 

In contrast to breastfeeding, the impact of COVID-19 restrictions on complementary feeding practices remains to be studied and reported in the literature. This survey study aimed to provide a better understanding of how COVID-19 restrictions affected complementary feeding practices among parents of infants aged 4 to 12 months during COVID-19 restrictions in Poland. 

## 2. Materials and Methods

### 2.1. Study Design 

This was a cross-sectional study involving a convenience series of participants, with the use of an online questionnaire. 

### 2.2. Questionnaire Design

The questionnaire was developed by the members of the study team from the Medical University of Warsaw, including two pediatric gastroenterologists with experience in infant nutrition research (AH, PD) and a dietitian (AS). The Nutricia Foundation, a non-commercial organization supporting nutrition research and education, was also involved. An online survey was chosen as the format due to the pandemic situation and the resultant COVID-19 restrictions. 

In brief, the questionnaire included general demographic data and questions pertaining to infant feeding in the context of COVID-19 restrictions. The questionnaire mostly included multiple choices or categorical response options. For some of the questions, more than one option could be chosen. The questionnaire needed approximately 15 min to complete. The questionnaire was accessible via a link that allowed an anonymous response. It was open during two intervals (27 May to 3 June 2020 and 17 March to 13 May 2021). The sample size was not formally calculated. During the first data collection (in 2020), access to the online questionnaire was disabled after receiving 5000 responses. 

The success with the recruitment stimulated us to repeat the survey (with three additional questions related to telehealth reflecting the changing healthcare environment). We planned to compare the responses during the two data collection periods, assuming it would allow us to analyze how COVID-19 restrictions impact complementary feeding practices. During the second data collection, the same number of responses was expected. However, the recruitment was unexpectedly slower, and access to the questionnaire was closed before the target sample size was reached. Additionally, whereas the study periods coincided with the timing of COVID-19 restrictions, neither of the study periods exactly coincided with the strict lockdown, which occasionally varied across the country. Considering the above, we abstained from comparing the two study periods. 

The validity of questionnaire was not tested, and a pilot survey was not performed.

### 2.3. Outcomes 

All outcomes were in the context of COVID-19 restrictions and included the following: sources of infant feeding-related information; impact of the COVID-19 pandemic on the method of feeding; changes in the feeding pattern due to the COVID-19 pandemic; impact of the COVID-19 pandemic on the introduction of complementary feeding; the use of food supplements (probiotics, prebiotics, synbiotics, and/or multivitamins) during the COVID-19 pandemic; the first point of contact for infant feeding problems during the COVID-19 pandemic; and most important infant feeding-related issues during the COVID-19 pandemic. Healthcare access and the use of telehealth during the COVID-19 pandemic, including the possibility of discussing infant feeding problems using a telehealth consultation, were assessed only in 2021, as the use of the telehealth was more common during the second survey interval. 

### 2.4. Participants 

Parents of infants aged 4 to 12 months (regardless of age, gender, race, ethnicity, level of education, financial situation, and place of residence) and those speaking Polish were eligible to participate in this study. The lower age limit was determined by the current European [10] and Polish [11] guidelines according to which complementary foods (solid and liquids other than breast milk or infant formula) should not be introduced before 4 months but should not be delayed in terms of introduction beyond 6 months. The upper age limit was determined by the fact that, at around 12 months of age, children are usually offered the same foods that the rest of the family eats. The recruitment was through parenting websites, blogs, community groups, social media, and word of mouth. There were no incentives for participation, and the respondents could discontinue the survey at any time.

### 2.5. Ethics 

Ethical approval was requested from the Medical University of Warsaw; however, it was judged not to be required. All participants voluntarily agreed to participate in the study and were informed that the data would be analyzed anonymously. All demographic data were recorded in broad categories only.

### 2.6. Statistical Analysis 

Data were analyzed by an independent statistician. Data for all outcomes are reported for all participants and, if feasible, for both years (2020 and 2021) jointly. Statistical analysis was conducted using the R package, version 4.0.5. Answers to questions are presented with total number of responses (n) and frequencies of subgroup (%). The relationship between nominal variables was evaluated by using the Chi-square test. Additionally, logistic regression analysis was carried out to evaluate the impact of certain sociodemographic characteristics (age of infant, education level of parents, place of residence, and financial situation) on the following: sources of information during COVID-19 restrictions (yes/no for use of selected sources); impact of COVID-19 restrictions on the method of feeding the child (impact/no impact); introduction of complementary feeding during COVID-19 restrictions (impact of pandemic/no impact); administration of food supplements to the child during COVID-19 restrictions (yes/no); impact of COVID-19 restrictions on access to doctors and the possibility of discussing infant feeding-related problems (impact/no impact); the use of telehealth to discuss the health of a child (yes/no); implementation of specific changes in the diet of a child during COVID-19 restrictions (yes/no for each kind of change); first point of contact when having problems with the feeding of a child during COVID-19 restrictions (yes/no for different points of contact); and key infant feeding-related issues during COVID-19 restrictions (yes/no for different issues). The phase of study (2020/2021) was included in each model as a covariate. All tests were two-sided with a significance level of α = 0.05.

## 3. Results

Data from the online questionnaire were available from 6934 responders (5000 responders during the first data collection and 1934 responders during the second data collection) who met the inclusion criteria.

### 3.1. Participants Characteristics and Survey Responses

#### 3.1.1. Participants’ Characteristics

Sociodemographic characteristics are shown in Table 1. The vast majority of responders (99.2%) were females, aged 26 to 30 years (47%). Most responders (75.5%) had infants aged 7–12 months. Most respondents (72%) declared having a tertiary level of education (i.e., they were subjects with university diplomas) and reported their family financial status as satisfactory (50.7%). Over half of the respondents (57.5%) declared no impact of COVID-19 on their financial situation; however, almost 40% reported that it rather changed or changed it for the worse (29% and 10.7%, respectively). 

#### 3.1.2. Survey Responses

Survey responses are shown in Table 2. 

*Sources of information.* The most frequently reported sources of infant-feeding related information were other parents, family members, or friends (48.6%); webinars and experts’ recommendations (40.8%); internet and apps (31.6%); and professional expert guides (27.3%). Not needing to deepen their knowledge of infant feeding was reported by 17.2% of participants. 

*Impact on the method of feeding.* Overall, significantly more participants reported that COVID-19 restrictions had ‘*no*’ or ‘*rather no’* impact on the method of feeding compared with those who reported ‘*yes*’ or ‘*rather yes*’ (87.7% versus 7.7%, *p* < 0.0001). 

*Changes in the feeding pattern.* Almost 12% of participants reported that their child received more homemade food; however, almost 8% reported that their child received more ready-to-eat food. Almost 7% reported that their child was more often breastfed.

*Complementary feeding.* Most participants (80.9%) reported that the COVID-19 pandemic had no impact on complementary feeding. 

*Supplement use.* Most participants (51.4%) reported regular (42.8%) or irregular (8.6%) use of food supplements (probiotics, prebiotics, synbiotics, and multivitamins). However, compared with the regular use of food supplements, no use was reported more frequently (42.8% versus 48.6%, *p* < 0.001). 

*First point of contact.* The first point of contact for infant feeding problems was a public (41.1%) or private (17.3%) healthcare system, including telehealth, followed by information obtained from websites (14.1%). 

*Most important infant feeding-related issues.* Most participants reported lack of access to doctors for consultation with regard to feeding advice (73.8%) and lack of access to fresh and safe foods (72.2%) as the largest infant feeding-related issues during the COVID-19 pandemic. 

*Healthcare access and telemedicine.* Three additional healthcare/telehealth-related questions were asked in 2021 only. Among 1934 participants, almost 70% felt that the COVID-19 pandemic impacted (‘*yes*’ or ‘*rather yes’*) their access to a physician to discuss infant feeding-related issues. 

Among those who used telehealth (*n* = 1474), when asked whether it provided the possibility of discussing infant nutrition problems, more responders answered ‘*no’* or *‘rather no’* compared with those who answered *‘yes’* or *‘rather yes’* (52.2% versus 47.8%, respectively; *p =* 0.02). 

### 3.2. Logistic Regression Analysis

Based on multivariate logistic regression (Table 3), the odds of parents using webinars/experts’ recommendations increased with infant age. These odds were also increased for parents with a tertiary vs. primary level of education and for parents living in cities above 20 k citizens vs. villages. However, the odds decreased for parents with the poorest financial situation (having money only for basic needs) vs. parents with the best financial situation (having enough money and some savings). The odds of parents using internet/apps as a source of knowledge were higher for parents with a tertiary vs. primary level of education and for parents living in a city (regardless of number of citizens) vs. a village. However, the odds of parents using internet/apps decreased for parents with the poorest financial situation (having money only for basic needs) vs. parents with the best financial situation (having enough money and some savings). The odds of parents using professional expert guides were higher for parents with younger infants and for parents with a secondary or tertiary vs. primary level of education, as well as among parents living in cities above 20 k citizens vs. villages. The odds of parents using professional expert guides were lower for almost every reported financial situation vs. parents with the best financial situation (having enough money and some savings). Parents with educational levels higher than a primary education were less likely to claim no need to deepen knowledge vs. parents with a primary education. The same was true for parents living in cities > 100 k citizens vs. villages. Receiving knowledge from other parents, family members, or friends was not significantly related to age of infant, education level of parents, or place of residence. However, the odds of parents obtaining knowledge from other parents, family and friends were higher for those from almost every financial situation group vs. parents with the best financial situation (having enough money and some savings).

Logistic regression on the feeding patterns and access to healthcare during COVID-19 restrictions was also carried out (Table 4). The odds for claiming that COVID-19 impacted the method of feeding the infant decreased with infant age and significantly increased for parents living in cities up to 20 k citizens or above 500 k citizens vs. villages, but the odds were not significantly associated with parent’s education level. These odds also increased for parents with average financial situations (having enough money but not enough for savings or living sparingly and, therefore, having enough money) vs. parents with the best financial situation (having enough money and some savings). Additionally, the odds for claiming that COVID-19 impacted the introduction of complementary feeding (extending the time, delaying, or stopping introduction) decreased with infant age, but the odds were not significantly associated with parent’s education level or location of residence. These odds also increased for parents with average financial situations (having enough money but not enough for savings, living sparingly and, therefore, having enough money, or living very sparingly to save money for more serious expenses) vs. parents with the best financial situation (having enough money and some savings). The odds of parents having the opinion that the pandemic impacted access to doctors regarding feeding-related problems of their child were higher for parents with a vocational or secondary vs. primary education level. The odds of parents using telehealth were higher for parents with older infants and for parents living in cities >500 k citizens vs. villages. The odds of parents giving food supplements to their child during COVID-19 restrictions were higher for older infants but were not significantly related to the parent’s education level, location of residence, or financial situation.

Separate logistic regression models were used to identify predictors of specific changes in the diets of children during COVID-19 restrictions (Table 5). With increases in infant age, there was an increase in the odds of a child receiving more new tastes/foods and more home-made food; however, these children also more often received less expensive food and less often received cow’s milk formula. The odds of a child receiving more ready-made food for children and more cow’s milk formula, as well as being more often breastfed, decreased with infant age. In terms of education, the odds of parents giving their child more new tastes/foods and cow’s milk formula more often decreased for parents with a tertiary vs. primary level of education. The odds of parents giving children less new tastes/foods and more ready-made food for children were higher for parents living in cities with 100–500 k citizens vs. those living in villages. In terms of financial situations, the odds of less often breastfeeding, more often feeding with cow’s milk formula, or giving more often less expensive food were higher for parents with the poorest financial situation (money only for basic needs) vs. parents with the best financial status (enough money and some savings). Additionally, the odds of more frequent breastfeeding or giving more foods from ecological farms were higher for parents with a slightly better financial situation (living very sparingly to save money for more serious expenses) vs. parents with the best financial status (enough money and some savings). Additionally, all financial situation groups as compared with parents with the best financial status (enough money and some savings) had higher odds of giving less expensive food more often.

Based on logistic regression (Table 6), the odds of a consultation with a doctor or at a hospital as the first point of contact in the case of infant feeding problems decreased with increasing infant age and were lower for parents living in cities with 20–100 k citizens vs. villages. Using websites as the first point of contact, if a feeding issue occurred, it was more likely occurring for parents living in cities with 20–100 k citizens compared to those living in villages. With increasing infant age, parents were more likely to claim they did not look for help in regard to feeding problems. Additionally, parents with an average financial situation (enough money but not enough for savings) were less likely to claim they did not seek any support compared to parents with the best financial status (enough money and some savings). 

The last group of logistic regression models was calculated with regard to major infant feeding-related issues encountered during COVID-19 restrictions (Table 7). The odds of issues with lack of access to a doctor for consultation about feeding issues decreased with infant age as well as for parents living in cities with more than 500 k citizens vs. villages. However, those odds increased for parents in almost every financial situation compared to parents with the best financial status (enough money and some savings). The issue of a lack of access to fresh and safe foods was less likely for parents living in cities over 500 k citizens vs. villages, and it was more likely for parents in some financial situations compared to parents with the best financial status (enough money and some savings). The odds of limited access to ready-to-eat food decreased with infant age. These odds also were lower for parents regardless of their education level compared to a primary education and were higher for parents in some financial situations compared to parents with the best financial status (enough money and some savings).

## 4. Discussion

### 4.1. Summary of Main Results

This online study carried out in Poland aimed to provide a better understanding of issues related to complementary feeding practices during COVID-19 restrictions among parents of infants aged 4 to 12 months. The survey involved more than 6900 participants, almost exclusively mothers. It was conducted twice, in 2020 and 2021, during the COVID-19 pandemic. However, none of the study periods exactly coincided with the strict lockdown, which occasionally varied across the country. As would be expected, the sources of information during COVID-19 varied. Most parents received information from more than one source, with other parents, family members, or friends being the most frequently reported. No major issues regarding the impact of COVID-19 restrictions on the method of feeding, changes in feeding patterns, or complementary feeding introduction were reported. Over half of the participants reported regular or irregular use of food supplements. The major issues reported by the participants were in regard to their access to healthcare providers and telemedicine consultations. More participants judged the latter as insufficient for discussing infant-feeding problems. 

The lack of impact of COVID-19 restrictions on complementary feeding practices may be associated with characteristics of the study population, which involved parents who had mostly a high (tertiary) level of education and/or a good financial situation (however, some reported that their financial situation changed for the worse). Financial status and its change due to the COVID-19 pandemic may have had an impact on some infant feeding practices, i.e., choices of cheaper foods and the consumption of more home-made foods. The impact of COVID-19 restrictions, i.e., loss of household income and/or limited access to fresh and safe foods, mostly affects the most vulnerable groups, including those of lower household income, Black and other ethnic minority groups, and/or parents with a low level of education [6]. Further studies assessing the way that child nutrition has changed in the groups most affected by COVID-19 restrictions are needed.

With regard to sources of information, multivariate logistic regression analysis showed most consistently that parents with a tertiary level of education and those living in a city above 500 k were at higher odds of using webinars/experts’ recommendations, internet/apps, and professional expert guides and lower odds of claiming not needing to deepen their knowledge. Parents from almost every financial situation group surveyed compared to parents with the best financial situation were less likely to use professional expert guides and were more likely to receive knowledge from other parents, family, and friends. With regard to feeding patterns, multivariate logistic regression analysis also showed that the only factors that predicted the impact of COVID-19 restrictions on the method of feeding were infant age and some financial situations. The odds of COVID-19 restrictions impacting method of feeding decreased with the infant age but increased for parents with average compared with the best financial situations. With regard to timing of complementary feeding, the odds for claiming that COVID-19 impacted factors (extending the time, delaying, or stopping) associated with the introduction of complementary feeding declined with infant age and were not significantly associated with parent’s education level or location of residence. However, the odds that the financial situation during COVID-19 restrictions impacted the timing of complementary feeding were increased for parents in almost every financial situation compared to those with the best financial situations. Logistic regression analysis also revealed changes in the infant diet during COVID-19 restrictions in Poland, which varied with regard to infant age, city of residence, level of education, and financial situation. Lastly, multivariate logistic regression analysis also showed differences among groups in the first point of contact when having problems with the feeding of a child during COVID-19 restrictions as well as with access to some healthcare or foods. With increasing infant age, parents were less likely to choose a doctor or hospital as the first point of contact or more likely to indicate no need for support (i.e., did not look for help). Parents with an average financial situation were also less likely to look for support. The odds of having issues with access to a doctor for consultation about feeding as a major feeding-related issue decreased with infant age and were lower for parents living in a city above 500 k; however, these odds were higher for parents in almost every financial situation group compared with the best financial situation.

### 4.2. Strengths and Limitations

The size of the study sample is the strength of this online survey. We used convenience sampling, i.e., the participants were self-selected through parenting websites, blogs, community groups, and social media. It is likely that those participants were more interested in infant feeding. Moreover, the study sample was skewed towards mothers with a tertiary level of education and, thus, potentially higher socio-economic status. The conclusions of our survey may not be applicable to mothers of lower socio-economic status. Lastly, those with no internet access, including potentially more disadvantaged individuals, were excluded from participating in this survey. Thus, sampling bias is possible. As with any cross-sectional study, our study is subject to non-response bias, i.e., the participants who agreed to contribute may differ from those who did not participate. Thus, the representativeness of the infant population may be questioned. There were almost no fathers among the responders. However, this finding reflects the real-world situation in Poland where care of infants and young children is usually undertaken by mothers. 

Finally, our decision to pool the data from two samples may be questioned. As stated earlier, the success with the recruitment stimulated us to repeat the survey (with three additional questions related to telehealth reflecting the changing healthcare environment). We planned to compare the responses during the two data collection periods, assuming it would allow us to analyze how COVID-19 restrictions impacted complementary feeding practices. We abstained from such a comparison for two reasons. First, surprisingly, the response rate during the second collection period was much slower, even though the same methodology was used. The reasons for this lower recruitment rate are unclear. However, the lack of willingness of parents to participate in this latter part of the study may have been due to the increasing number of online surveys during the pandemic. Second, it became clear that the restrictions differed both throughout the country and with the phase of the pandemic, and the survey did not allow us to collect information on the type of such restrictions. 

### 4.3. Agreement and Disagreement with Other Studies

*Sources of information.* Sources of information may vary depending on the study location and timing of assessment, i.e., before or during COVID-19 restrictions. For the latter, a 2021 COVID-19 New Mum Study (UK) assessed the impact of COVID-19 on the experiences and feeding practices of new mothers [8]. This study found that the most frequently reported sources of information included a partner (60%), health professionals (50%), or an online support group (47%). Similarly, in our study, the most frequently reported sources were other parents, family members or friends (48.6%) as well as webinars and expert recommendations (40.8%). Regarding the pre-pandemic era, a 2020 cross-sectional, very small study conducted in Polish lactating women (*n* = 33) reported that 85% of participants used the internet as a main source of information on lactation and infant feeding. In contrast to our study, friends with children were a less frequent (36%) choice [12]. 

*Impact on infant feeding practices.* We found that COVID-19 restrictions largely had no impact on feeding practices in Poland, including the introduction of complementary feeding. However, multivariate logistic regression analysis did reveal that infants of parents with average compared with the best financial situations during COVID-19 restrictions were more likely experience impacted timing of complementary feeding. In the COVID-19 New Mum Study (UK), only 13% of women (*n* = 177 of 1365) reported changes in infant feeding associated with the COVID-19 pandemic [9]. However, 57% (*n* = 601 of 1049) of women who delivered before lockdown reported that their feeding support decreased since lockdown. In the same study, 45% (141 of 316) of women who delivered during lockdown declared that did not receive enough feeding support (except hospital assistance). Insufficient access to physicians regarding nutritional advice during the COVID-19 pandemic was also an issue in our study, particularly for parents with only average compared with the best financial situations.

*Food supplements.* Healthy infants consuming a well-balanced diet do not need any food supplements. However, our study found interest in food supplements among the majority of responders, as most of the infants received food supplements such as probiotics, prebiotics, symbiotic, and/or multivitamins either regularly or irregularly. Regretfully, information on the exact type(s) of food supplements was not obtained. One can only speculate that the overuse of food supplements was seen as a means of improving immunity. During the COVID-19 pandemic, there has been interest in using food supplements as a potential strategy for supporting immunity and reducing the risk of respiratory infections, especially in adults, but evidence from large, randomized controlled trials is lacking. 

*Access to healthcare and telehealth.* We are not aware of any studies that have systematically monitored delays in access to healthcare associated with the pandemic. However, several studies have documented reduced pediatric consultations because of COVID-19 restrictions [13,14]. The current study documented concerns with regard to telehealth. However, telehealth is globally unlikely to disappear. Opportunities and challenges for telehealth within and beyond the COVID-19 pandemic have been discussed elsewhere [15,16]. A 2020 systematic review found that the use of telehealth during the COVID-19 pandemic improves the provision of health services [17]. While in-person pediatric consultations are invaluable, telehealth changes how care and education can be delivered. One such positive example is the role of telehealth in promoting breastfeeding [18]. In our opinion, telehealth, if accessible to all parents, provides an opportunity to promote appropriate infant feeding practices, and it is likely to help in improving the health and development of children.

## 5. Conclusions

There have been several studies on the impact of COVID-19 restrictions on breastfeeding. However, the impact of COVID-19 restrictions on complementary feeding practices remains to be studied and reported in the literature. The results of this online cross-sectional survey clarify the major issues associated with complementary feeding practices during the implementation of COVID-19 restrictions in Poland. These findings apply to the population studied. Further studies in more vulnerable groups are needed.

## Figures and Tables

**Table 1 nutrients-13-03196-t001:** Sociodemographic characteristics of survey participants, *n* (%).

Total	6934 (100)
**Gender**	
Female	6880 (99.2)
Male	54 (0.8)
**Age, responders**	
Up to 25 y	1132 (16.3)
26–30 y	3267 (47.0)
31–35 y	2066 (29.8)
>36 y	469 (6.8)
**Age, infants**	
4–6 months	1695 (24.4)
7–12 months	5239 (75.5)
**Education**	
Elementary (primary)	41 (0.6)
Secondary	654 (9.4)
Vocational	1197 (17.3)
Tertiary (subjects with university diplomas)	4995 (72.0)
None declared	47 (0.7)
**Financial family status (perceived by parents)**	
Enough money and some saving	3519 (50.7)
Enough money but not enough for savings	2144 (30.9)
Living sparingly and, therefore, have enough money	999 (14.4)
Living very sparingly to save money for more serious expenses	136 (2.0)
Money only for basic needs	136 (2.0)
**Impact of COVID-19 pandemic on family material (financial) situation**	
Change for the better	59 (0.85)
Rather change for the better	135 (1.95)
No change	3984 (57.5)
Rather change for the worse	2012 (29.0)
Change for the worse	744 (10.7)
**City of residence**	
Village (rural area)	1862 (26.85)
City of 20 k citizens	746 (10.8)
City of 20–100 k citizens	1382 (19.9)
City of 100–500 k citizens	1224 (17.65)
City of above 500 k citizens	1720 (24.8)

**Table 2 nutrients-13-03196-t002:** Summary of the responses (*n* = 6934).

Sources of Information during the COVID-19 Pandemic	N (%)
Other parents, family members, or friends	3368 (48.6)
Webinars/experts’ recommendations	2832 (40.8)
Internet and apps	2192 (31.6)
Professional expert guides	1896 (27.3)
No need to deepen the knowledge of feeding	1195 (17.2)
Other sources	291 (4.2)
**Did the COVID-19 pandemic impact the method of feeding your child?**	
Yes	146 (2.1)
Rather yes	385 (5.6)
Neutral	323 (4.7)
Rather no	1877 (27.1)
No	4203 (60.6)
**What has changed in the diet of your child during the COVID-19 pandemic?**	
My child is currently fed by someone else	14 (0.2)
My child received less new tastes/foods	386 (5.6)
My child receives more new tastes/foods	386 (5.6)
My child receives more home-made food	806 (11.6)
My child receives more ready-made food for children	548 (7.9)
My child receives more often less expensive foods	150 (2.2)
My child receives more often more expensive foods	167 (2.4)
My child receives more foods from ecological farms (‘bio/eco’)	336 (4.8)
My child is more often breastfed	481 (6.9)
My child is less often breastfed	108 (1.6)
My child receives more often cow’s milk formula	121 (1.7)
My child received less often cow’s milk formula	143 (2.1)
**Complementary feeding introduction during the COVID-19 pandemic**	
Pandemic did not impact the method of feeding and introducing new flavors	5613 (80.9)
I introduce a majority of products, with extension of time between new flavors/meals	485 (7.0)
I introduce a majority of products, delaying introduction of potentially allergenic products as having higher risk of allergy (e.g., eggs, fish, and wheat)	187 (2.7)
I am extending the time between new meals/flavors, especially potentially allergenic products	85 (1.2)
I do not introduce new products as I am afraid of intolerance	25 (0.4)
Did your child received food supplements during the COVID-19 pandemic?	
No	3370 (48.6)
Irregularly	594 (8.6)
Yes	2970 (42.8)
**What was your first point of contact when having problems with the feeding of your child during the COVID-19 pandemic?**	
Public healthcare doctor (including telehealth)	2850 (41.1)
Private visit to a doctor	1200 (17.3)
Websites	980 (14.1)
More experienced parents, friends, and family members	920 (13.3)
Did not look for help	755 (10.9)
Other sources	194 (2.8)
Hospital	35 (0.5)
**The most important infant feeding-related issues during the COVID-19 pandemic**	
Lack of access to doctors for consultation about feeding	5116 (73.8)
Lack of access to fresh and safe foods	5003 (72.2)
Limited access to ready-made foods for children	1432 (20.7)
Other	3073 (44.3)
**Do you think that the COVID-19 pandemic impacted access to doctors and the possibility to discuss infant- feeding related problems? ^1^**	
Yes	859 (44.4)
Rather yes	490 (25.3)
Rather no	362 (18.7)
No	223 (11.5)
**Did you use telehealth at least once to discuss the health of your child? ^1^**	
No	460 (23.8)
Yes	1474 (76.2)
**Do you think that telehealth gives you the possibility to discuss nutrition problems of your child? ^1,2^**	
Yes	123 (8.3)
Rather yes	582 (39.5)
Rather no	429 (29.1)
No	340 (23.1)

^1^ Questions asked in 2021 only; *n* = 1934 responders.^2^ Calculated in relation to *n* = 1474 responders using telehealth.

**Table 3 nutrients-13-03196-t003:** Logistic regression on the different sources of knowledge during COVID-19 restrictions.

Predictor	Other Parents, Family, and Friends		Webinars/Experts’ Recommendation		Internet/Apps		Professional Expert Guides		No Need to Deepen Knowledge	
	OR (95% CI)	*p*	OR (95% CI)	*p*	OR (95% CI)	*p*	OR (95% CI)	*p*	OR (95% CI)	*p*
**Age, infants, and months**										
	0.99 (0.98;1.02)	0.946	1.03 (1.01;1.05)	0.005	1.01 (0.99;1.03)	0.504	0.97 (0.95;0.99)	0.010	1.01 (0.98;1.03)	0.948
**Education (baseline = primary)**										
Vocational	0.99 (0.50;1.95)	0.972	1.28 (0.49;3.99)	0.631	1.67 (0.69;4.69)	0.289	2.08 (0.68;9.06)	0.251	0.48 (0.24;0.97)	0.039
Secondary	1.06 (0.57;1.98)	0.862	2.43 (1.03;7.13)	0.066	2.30 (1.02;6.17)	0.066	3.77 (1.35;15.70)	0.028	0.37 (0.20;0.71)	0.002
Tertiary	1.03 (0.55;1.92)	0.929	4.15 (1.77;12.13)	0.003	3.98 (1.77;10.65)	0.002	5.44 (1.95;22.60)	0.005	0.22 (0.12;0.42)	<0.001
**City of residence (baseline = village)**										
City (up to 20 k)	0.99 (0.83;1.17)	0.898	0.93 (0.76;1.12)	0.440	1.31 (1.08;1.58)	0.006	1.11 (0.90;1.36)	0.312	0.90 (0.72;1.11)	0.321
City (20–100 k)	0.96 (0.84;1.11)	0.578	1.21 (1.04;1.42)	0.013	1.23 (1.05;1.44)	0.010	1.19 (1.01;1.40)	0.041	0.87 (0.73;1.04)	0.137
City (100–500 k)	0.97 (0.84;1.13)	0.717	1.38 (1.18;1.61)	<0.001	1.24 (1.06;1.46)	0.009	1.44 (1.22;1.70)	<0.001	0.82 (0.67;0.99)	0.038
City (above 500 k)	0.93 (0.81;1.06)	0.286	1.50 (1.30;1.73)	<0.001	1.60 (1.38;1.86)	<0.001	1.67 (1.44;1.95)	<0.001	0.67 (0.55;0.80)	<0.001
**Financial situation (baseline = enough money and some savings)**										
Enough money but not enough for savings	1.13 (1.01;1.26)	0.032	1.03 (0.91;1.15)	0.658	1.05 (0.93;1.18)	0.448	0.83 (0.73;0.94)	0.003	0.91 (0.79;1.05)	0.213
Living sparingly and, therefore, have enough money	1.20 (1.04;1.39)	0.012	0.90 (0.77;1.06)	0.204	1.13 (0.97;1.33)	0.116	0.85 (0.72;0.99)	0.046	0.86 (0.71;1.04)	0.128
Living very sparingly to save money for more serious expenses	1.10 (0.78;1.55)	0.604	1.07 (0.73;1.55)	0.730	0.87 (0.58;1.28)	0.491	0.91 (0.60;1.35)	0.660	1.03 (0.65;1.57)	0.903
Money only for basic needs	1.43 (1.01;2.03)	0.044	0.60 (0.38;0.92)	0.023	0.59 (0.37;0.90)	0.017	0.45 (0.26;0.72)	0.002	1.24 (0.81;1.86)	0.305

A separate model was prepared for each source of knowledge as the dependent variable, and predictors in each model included the following: age of infant, parent’s education level, place of residence, and financial situation. OR—odds ratio with 95% confidence interval (CI).

**Table 4 nutrients-13-03196-t004:** Logistic regression on the feeding patterns and access to healthcare during COVID-19 restrictions.

Predictor	COVID-19 Impact on the Method of Feeding		COVID-19 Impact on Introduction of Complementary Feeding		COVID-19 Impact on Access to Doctors Refeeding-Related Problems		Receival of Food Supplements during COVID-19 Restrictions		Use of Telehealth to Discuss Health of Child	
	OR (95% CI)	*p*	OR (95% CI)	*p*	OR (95% CI)	*p*	OR (95% CI)	*p*	OR (95% CI)	*p*
**Age, infants, and months**										
	0.96 (0.92;0.99)	0.028	0.92 (0.89;0.95)	<0.001	1.04 (0.99;1.09)	0.072	1.03 (1.01;1.05)	0.004	1.07 (1.02;1.12)	0.005
**Education (baseline = primary)**										
Vocational	0.98 (0.30;4.41)	0.974	0.89 (0.36;2.55)	0.814	3.15 (1.15;8.67)	0.025	0.94 (0.48;1.86)	0.857	1.32 (0.42;3.76)	0.608
Secondary	0.88 (0.31;3.70)	0.836	0.80 (0.36;2.16)	0.630	2.64 (1.06;6.61)	0.035	1.11 (0.59;2.09)	0.739	1.15 (0.40;2.96)	0.786
Tertiary	0.96 (0.34;3.99)	0.942	0.79 (0.35;2.11)	0.602	2.21 (0.89;5.48)	0.083	1.24 (0.66;2.31)	0.501	1.51 (0.52;3.89)	0.409
**City of residence (baseline = village)**										
City (up to 20 k)	1.53 (1.08;2.14)	0.015	1.19 (0.92;1.54)	0.169	1.29 (0.90;1.86)	0.172	0.99 (0.83;1.17)	0.977	1.16 (0.80;1.70)	0.449
City (20–100 k)	1.27 (0.94;1.70)	0.119	0.86 (0.69;1.08)	0.202	1.07 (0.81;1.41)	0.640	1.01 (0.88;1.16)	0.888	1.00 (0.74;1.34)	0.987
City (100–500 k)	1.13 (0.82;1.55)	0.463	0.99 (0.78;1.25)	0.934	1.09 (0.81;1.47)	0.569	1.11 (0.96;1.29)	0.869	1.01 (0.74;1.39)	0.951
City (above 500 k)	1.38 (1.04;1.84)	0.026	0.94 (0.76;1.17)	0.607	1.11 (0.84;1.47)	0.475	0.99 (0.86;1.13)	0.158	1.48 (1.08;2.05)	0.016
**Financial situation (baseline = enough money and some savings)**										
Enough money but not enough for savings	1.27 (1.01;1.60)	0.041	1.20 (1.01;1.43)	0.044	1.06 (0.84;1.32)	0.632	0.99 (0.86;1.13)	0.910	1.06 (0.83;1.35)	0.654
Living sparingly and, therefore, have enough money	1.73 (1.31;2.27)	<0.001	1.93 (1.57;2.37)	<0.001	1.20 (0.89;1.62)	0.247	1.06 (0.92;1.22)	0.446	1.39 (1.01;1.95)	0.053
Living very sparingly to save money for more serious expenses	1.47 (0.71;2.73)	0.259	1.72 (1.02;2.78)	0.033	1.45 (0.77;2.90)	0.264	1.22 (0.86;1.73)	0.267	1.83 (0.91;4.10)	0.112
Money only for basic needs	1.51 (0.73;2.81)	0.229	1.64 (0.97;2.64)	0.052	1.54 (0.85;2.96)	0.174	1.13 (0.80;1.60)	0.497	1.02 (0.58;1.90)	0.937

A separate model was prepared for each source of knowledge as the dependent variable, and the predictors in each model included the following: age of infant, parent’s education level, place of residence, and financial situation. OR—odds ratio with 95% confidence interval (CI).

**Table 5 nutrients-13-03196-t005:** Logistic regression on changes in the diet of infant during COVID-19 restrictions.

Predictor	My Infant Fed by Someone Else		My Infant Receives Less New Tastes/Foods		My Infant Receives More New Tastes/Foods		My Infant Receives More Homemade Food	
	OR (95% CI)	*p*	OR (95% CI)	*p*	OR (95% CI)	*p*	OR (95% CI)	*p*
**Age, infants, and months**								
	1.05 (0.84;1.33)	0.694	1.12 (0.97;1.17)	0.245	1.13 (1.08;1.18)	<0.001	1.23 (1.19;1.28)	<0.001
**Education (baseline=primary)**								
Vocational	2.09 (0.06;72.17)	>0.999	0.96 (0.14;19.29)	0.971	0.69 (0.20;2.80)	0.576	1.25 (0.37;5.01)	0.735
Secondary	1.68 (0.01;20.54)	0.998	2.21 (0.43;40.57)	0.449	0.39 (0.13;1.47)	0.125	0.96 (0.31;3.60)	0.940
Tertiary	1.99 (0.03;63.69)	0.998	2.65 (0.52;48.61)	0.349	0.28 (0.09;1.06)	0.037	0.80 (0.26;2.99)	0.709
**City of residence (baseline=village)**								
City (up to 20 k)	4.42 (0.42;95.69)	0.226	1.21 (0.84;1.71)	0.301	0.77 (0.50;1.14)	0.195	1.16 (0.86;1.55)	0.324
City (20–100 k)	1.42 (0.06;35.96)	0.805	0.99 (0.73;1.35)	0.955	0.89 (0.64;1.23)	0.469	0.96 (0.75;1.23)	0.738
City (100–500 k)	5.70 (0.83;11.25)	0.122	1.35 (1.01;1.81)	0.043	1.06 (0.77;1.46)	0.719	0.81 (0.63;1.04)	0.105
City (above 500 k)	6.03 (0.98;11.61)	0.102	1.30 (0.99;1.71)	0.063	0.88 (0.64;1.20)	0.427	0.85 (0.67;1.08)	0.181
**Financial situation (baseline=enough money and some savings)**								
Enough money but not enough for savings	0.82 (0.25;2.42)	0.729	1.07 (0.86;1.33)	0.559	0.72 (0.55;0.93)	0.011	0.92 (0.76;1.12)	0.412
Living sparingly and, therefore, have enough money	0.81 (0.01;57.03)	0.989	1.16 (0.88;1.52)	0.286	0.88 (0.65;1.20)	0.444	1.22 (0.96;1.54)	0.099
Living very sparingly to save money for more serious expenses	0.83 (0.01;26.57)	0.996	1.02 (0.48;1.97)	0.955	0.57 (0.22;1.25)	0.202	1.26 (0.71;2.17)	0.417
Money only for basic needs	0.83 (0.01;15.23)	0.996	0.43 (0.13;1.07)	0.107	1.26 (0.62;2.41)	0.498	1.50 (0.83;2.66)	0.166
**Predictor**	**My Infant Receives More Ready-made Food for Children**		**My Infant Receives more often Less Expensive Food**		**My Infant Receives More often More Expensive Food**		**My Infant Receives More Foods from Ecological Farms**	
	**OR (95% CI)**	** *p* **	**OR (95% CI)**	** *p* **	**OR (95% CI)**	** *p* **	**OR (95% CI)**	** *p* **
**Age, infants, and months**								
	0.96 (0.92;0.99)	0.036	1.17 (1.09;1.26)	<0.001	1.03 (0.97;1.10)	0.363	1.02 (0.97;1.07)	0.437
**Education (baseline=primary)**								
Vocational	0.62 (0.18;2.52)	0.469	0.68 (0.09;14.09)	0.739	0.32 (0.05;2.50)	0.216	0.75 (0.16;5.44)	0.742
Secondary	0.51 (0.17;1.92)	0.271	0.86 (0.15;16.57)	0.892	0.32 (0.08;2.15)	0.153	0.76 (0.19;5.02)	0.728
Tertiary	0.57 (0.19;2.12)	0.355	0.72 (0.13;13.65)	0.758	0.39 (0.10;2.55)	0.227	0.79 (0.21;5.16)	0.762
**City of residence (baseline=village)**								
City (up to 20 k)	1.18 (0.83;1.66)	0.349	1.32 (0.73;2.33)	0.344	1.55 (0.89;2.66)	0.382	0.89 (0.57;1.36)	0.586
City (20–100 k)	1.20 (0.90;1.60)	0.206	0.92 (0.55;1.54)	0.761	1.02 (0.60;1.70)	0.116	1.06 (0.75;1.50)	0.745
City (100–500 k)	1.50 (1.13;1.99)	0.005	1.12 (0.67;1.86)	0.661	1.25 (0.76;2.05)	0.953	1.09 (0.77;1.54)	0.621
City (above 500 k)	1.21 (0.92;1.58)	0.181	1.15 (0.69;1.90)	0.600	1.35 (0.86;2.14)	0.373	1.02 (0.74;1.42)	0.896
**Financial situation (baseline=enough money and some savings)**								
Enough money but not enough for savings	1.15 (0.93;1.43)	0.192	2.34 (1.43;3.91)	0.001	1.20 (0.85;1.71)	0.301	1.08 (0.83;1.39)	0.563
Living sparingly and, therefore, have enough money	1.22 (0.93;1.43)	0.150	7.41 (4.63;12.17)	<0.001	0.88 (0.53;1.41)	0.599	0.80 (0.56;1.13)	0.216
Living very sparingly to save money for more serious expenses	1.12 (0.56;2.07)	0.735	11.53 (5.25;24.24)	<0.001	0.88 (0.21;2.46)	0.827	1.09 (0.01;0.50)	0.029
Money only for basic needs	0.52 (0.20;1.16)	0.144	7.75 (3.07;17.84)	<0.001	0.92 (0.22;2.64)	0.898	0.37 (0.09;1.05)	0.105
**Predictor**	**My infant is more often breastfed**		**My infant is less often breastfed**		**My infant receives more often cow’s milk formula**		**My infant receives less often cow’s milk formula**	
	**OR (95% CI)**	** *p* **	**OR (95% CI)**	** *p* **	**OR (95% CI)**	** *p* **	**OR (95% CI)**	** *p* **
**Age, infants, and months**								
	0.94 (0.90;0.98)	0.004	1.01 (0.93;1.09)	0.786	0.88 (0.82;0.95)	0.001	1.13 (1.05;1.22)	0.001
**Education (baseline = primary)**								
Vocational	1.34 (0.21;26.15)	0.794	1.46 (0.04;12.62)	0.975	0.27 (0.06;1.46)	0.099	5.27 (0.01;1.22)	0.972
Secondary	2.03 (0.38;37.60)	0.504	2.16 (0.02;87.66)	0.974	0.26 (0.07;1.23)	0.054	4.07 (0.03;60.75)	0.973
Tertiary	3.07 (0.59;56.70)	0.286	2.61 (0.02;81.56)	0.974	0.15 (0.04;0.68)	0.005	2.19 (0.01;75.43)	0.974
**City of residence (baseline=village)**								
City (up to 20 k)	0.98 (0.68;1.40)	0.934	0.88 (0.40;1.78)	0.738	1.54 (0.83;2.78)	0.161	0.60 (0.29;1.14)	0.137
City (20–100 k)	0.92 (0.68;1.24)	0.568	1.20 (0.69;2.09)	0.518	1.31 (0.76;2.24)	0.323	1.03 (0.63;1.65)	0.913
City (100–500 k)	0.88 (0.65;1.20)	0.419	0.95 (0.52;1.71)	0.867	1.17 (0.66;2.05)	0.596	1.05 (0.64;1.70)	0.839
City (above 500 k)	1.01 (0.76;1.33)	0.992	0.88 (0.49;1.54)	0.649	0.87 (0.48;1.57)	0.654	0.71 (0.42;1.19)	0.197
**Financial situation (baseline=enough money and some savings)**								
Enough money but not enough for savings	0.91 (0.72;1.15)	0.445	1.42 (0.91;2.22)	0.116	1.35 (0.87;2.08)	0.173	0.86 (0.58;1.28)	0.464
Living sparingly and, therefore, have enough money	1.07 (0.80;1.42)	0.643	1.02 (0.54;1.83)	0.946	1.41 (0.83;2.33)	0.196	0.87 (0.53;1.39)	0.573
Living very sparingly to save money for more serious expenses	2.81 (1.60;4.84)	<0.001	2.21 (0.65;5.78)	0.143	0.57 (0.01;42.57)	0.976	0.26 (0.01;1.23)	0.188
Money only for basic needs	1.30 (0.62;2.52)	0.464	4.22 (1.53;9.96)	0.002	2.87 (1.10;6.62)	0.019	2.09 (0.81;4.66)	0.093

A separate model was prepared for each change in the diet of child as the dependent variable, and the predictors in each model included the following: age of infant, parent’s education level, place of residence, and financial situation. OR—odds ratio with 95% confidence interval (CI).

**Table 6 nutrients-13-03196-t006:** Logistic regression on first point of contact when having problems with the feeding of a child during COVID-19 restrictions.

Predictor	Doctor/Hospital		Websites		Other Parents, Friends, and Family Members		Did not Look for Help	
	OR (95% CI)	*p*	OR (95% CI)	*p*	OR (95% CI)	*p*	OR (95% CI)	*p*
**Age, infants, and months**								
	0.98 (0.96;0.99)	0.022	1.02 (0.99;1.05)	0.109	1.00 (0.94;1.29)	0.849	1.07 (1.03;1.11)	0.002
**Education (baseline = primary)**								
Vocational	1.21 (0.54;2.26)	0.752	3.31 (0.27;20.57)	0.955	0.93 (0.35;2.90)	0.885	0.27 (0.05;1.19)	0.094
Secondary	0.86 (0.43;1.62)	0.642	6.71 (0.29;25.46)	0.953	0.99 (0.42;2.90)	0.979	0.35 (0.07;1.19)	0.145
Tertiary	0.72 (0.36;1.36)	0.321	1.03 (0.44;3.69)	0.952	1.11 (0.47;3.25)	0.827	0.29 (0.06;1.11)	0.082
**City of residence (baseline = village)**								
City (up to 20 k)	0.88 (0.74;1.04)	0.135	1.17 (0.91;1.50)	0.221	1.01 (0.78;1.29)	0.962	1.25 (0.87;1.79)	0.233
City (20–100 k)	0.85 (0.74;1.04)	0.028	1.32 (1.08;1.62)	0.007	1.03 (0.84;1.27)	0.743	1.23 (0.87;1.79)	0.193
City (100–500 k)	0.93 (0.81;1.08)	0.371	1.22 (0.98;1.51)	0.062	1.01 (0.81;1.25)	0.933	1.03 (0.75;1.41)	0.848
City (above 500 k)	0.95 (0.83;1.09)	0.498	1.22 (0.99;1.51)	0.216	1.04 (0.85;1.26)	0.724	0.87 (0.64;1.17)	0.349
**Financial situation (baseline = enough money and some savings)**								
Enough money but not enough for savings	1.06 (0.95;1.18)	0.303	0.99 (0.85;1.16)	0.957	0.95 (0.81;1.12)	0.558	0.76 (0.60;0.97)	0.029
Living sparingly and, therefore, have enough money	0.92 (0.80;1.06)	0.262	1.10 (0.89;1.34)	0.336	1.09 (0.88;1.33)	0.427	1.08 (0.80;1.47)	0.611
Living very sparingly to save money for more serious expenses	0.89 (0.63;1.27)	0.526	0.92 (0.52;1.52)	0.747	1.10 (0.65;1.76)	0.717	10.39 (0.62;3.08)	0.415
Money only for basic needs	0.84 (0.59;1.20)	0.339	0.63 (0.32;1.14)	0.157	1.04 (0.61;1.69)	0.873	1.47 (0.65;3.31)	0.349

A separate model was prepared for each point of contact as the dependent variable, and the predictors in each model included the following: age of infant, parent’s education level, place of residence, and financial situation. OR—odds ratio with 95% confidence interval (CI).

**Table 7 nutrients-13-03196-t007:** Logistic regression on key infant feeding-related issues during COVID-19 restrictions.

Predictor	Lack of Access to Doctor for Consultation about Feeding		Lack of Access to Fresh and Safe Foods		Limited Access to Ready-to-Eat Foods for Children	
	OR (95% CI)	*p*	OR (95% CI)	*p*	OR (95% CI)	*p*
**Age, infants, and months**						
	0.93 (0.91;0.95)	<0.001	0.99 (0.97;1.01)	0.500	0.92 (0.89;0.94)	<0.001
**Education (baseline = primary)**						
Vocational	1.59 (0.68;3.51)	0.261	1.22 (0.53;2.66)	0.621	0.40 (0.20;0.81)	0.009
Secondary	1.09 (0.50;2.18)	0.819	0.84 (0.39;1.69)	0.643	0.31 (0.17;0.59)	<0.001
Tertiary	0.90 (0.42;1.80)	0.785	0.73 (0.34;1.46)	0.399	0.25 (0.13;0.47)	<0.001
**City of residence (baseline = village)**						
City (up to 20 k)	0.94 (0.77;1.15)	0.541	1.08 (0.89;1.31)	0.457	1.10 (0.89;1.35)	0.371
City (20–100 k)	0.99 (0.84;1.17)	0.921	1.10 (0.94;1.29)	0.250	1.11 (0.93;1.31)	0.246
City (100–500 k)	0.87 (0.73;1.02)	0.094	0.93 (0.79;1.09)	0.358	0.93 (0.77;1.12)	0.437
City (above 500 k)	0.77 (0.66;0.89)	0.001	0.83 (0.72;0.96)	0.015	0.88 (0.74;1.04)	0.128
**Financial situation (baseline = enough money and some savings)**						
Enough money but not enough for savings	1.27 (1.12;1.44)	<0.001	1.07 (0.95;1.21)	0.276	1.06 (0.92;1.22)	0.401
Living sparingly and, therefore, have enough money	1.48 (1.25;1.76)	<0.001	1.66 (1.40;1.98)	<0.001	1.33 (1.12;1.58)	0.001
Living very sparingly to save money for more serious expenses	1.35 (0.90;2.09)	0.153	1.37 (0.92;2.10)	0.127	1.32 (0.87;1.95)	0.175
Money only for basic needs	1.96 (1.25;3.19)	0.005	1.82 (1.19;2.88)	0.008	2.05 (1.40;2.96)	<0.001

A separate model was prepared for each infant feeding-related issue as the dependent variable, and the predictors in each model included the following: age of infant, parent’s education level, place of residence, and financial situation. OR—odds ratio with 95% confidence interval (CI).

## Data Availability

The dataset used and/or generated during this study is available from a given author upon reasonable request.

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
