# Peer review of "An Online Cross-Sectional Survey of Complementary Feeding Practices during the COVID-19 Restrictions in Poland"

_nutrients, 2021, doi:10.3390/nu13093196_

Round 1

Reviewer 1 Report

SPECIFIC COMMENTS:

ABSTRACT

The abstract should be rewritten based on the revisions applied to the text (as suggested below).

INTRODUCTION

The introduction fails to provide a useful background for the study. Given the declared aim of the study, a more extensive description of the restrictive measures imposed to the polish population during the COVID-19 pandemic would seem appropriate (and interesting), together with a proposed explanation as to why and how such measures could impact infant feeding practices. Likewise, the term “telehealth” and the extent and modality of its implementation should be described in greater detail.

Moreover, “infant feeding practices” is a rather broad concept, especially when considering the age range of the infants included in the study, and should be further explained.  

Additionally, an adequate and relevant literature review, useful to “set the ground” and explain why the present study was “necessary”, is lacking.

MATERIALS AND METHODS

The authors should include enough concise information in the methods section that it would be reproducible by other researchers.

Questionnaire design: A more detailed description of how the questionnaire was created would be of interest to the readers. What are the specific competences of the team who developed the questionnaire? It would be interesting to know if and how the authors tested content validity. Furthermore, I wonder if the survey was pilot-tested before the beginning of the present study in order to assess understanding of the questions (if so, please describe).

It is not clear why the questionnaire was proposed during two time periods. What was the difference between the two periods (if any)? What was the aim of said distinction if no comparison between the two “phases” was made? What restrictive measures were applied during the two time periods? Please explain.

It is not clear why the first period ended at 5000 responses, while the second one at way less (although remaining open for a longer period of time: 17 March to 13 May 2021 vs. 27 May to 3 June 2020). Please explain.

Outcomes: definitions of the outcomes are needed. What does the sentence “All outcomes were in the context of the COVID-19 restrictions” mean? Please explain.

Participants: on which grounds was the age range (4-12 months) set? Please explain.

Results: “Data from the online questionnaire were available from 6934 responders…who met the inclusion criteria”: it is not clear which exactly were the inclusion criteria. Were there any exclusion criteria? Please explain.

Participants characteristics: what does “tertiary level of education” mean? Please explain. What is the significance of the question on the impact of COVID-19 on participants’ financial situation? How does it fit with the declared aim of the study?

Survey responses: a bulleted list does not seem to be the best way to describe study results in a scientific publication. I would suggest you rewrite these results in a more discursive manner.

Why were the three healthcare/telehealth-related questions added in the second time period only? What changed between the first and the second period? Was telehealth not implemented during the first time period? Please explain.

Tables: what do “vocational” and “answer denial” mean? Please find better suited synonyms.  

Discussion – Summary of main results: Reference to the data currently available in the literature is interesting, even if not always appropriate or well explained.

Sources of information: the literature review seems a bit off-topic with respects to the declared aim of the study and the conclusion that “Overall, both pandemic and pre-pandemic data document that there is a need to increase parents’ eagerness to look for professional support (i.e., pediatrician, dietitian, midwifery)” is beyond the scope of the present study. I would suggest a more focused literature review and comparison with the present study findings.

Impact of (--> on) infant feeding practices: since this is one of the main outcomes of the study, a more detailed literature review would seem appropriate. The only study referenced is a 2020 UK paper by Vazquez-Vazquez et al., which the present study seems largely inspired to.

Food supplements: the sentence “However, a recent large observational study, which included data from more than 400,000 COVID-19 Symptom Study app users, found that women taking multivitamins, omega-3 fatty acids, probiotics, or vitamin D were less likely to test positive for SARS-CoV-2. Randomized controlled trials to assess the efficacy of selected food supplements for preventing COVID-19 are needed” is completely off-topic. I would suggest you remove it.

Conclusions: the conclusions should be rewritten. In their current form they do not offer a clear conclusion to the study, nor they reflect the study findings. Since the Authors found that “No major issues regarding the impact of the COVID-19 restrictions on the way of feeding, changes in feeding patterns, or complementary feeding introduction were reported”, it is not clear how they can conclude that “improving understanding of how the COVID-19 restrictions impact infant feeding practices could be of value when planning further quality pediatric healthcare”.

Additional considerations: some minor mistakes are present, and the paper would benefit from being revised and possibly copyedited by a native English speaker, to improve its readability. In particular, the way dates are written should be revised.

Author Response

ABSTRACT

The abstract should be rewritten based on the revisions applied to the text (as suggested below).                                                                                                                                                           

RESPONSE: Done.

INTRODUCTION

The introduction fails to provide a useful background for the study. Given the declared aim of the study, a more extensive description of the restrictive measures imposed to the polish population during the COVID-19 pandemic would seem appropriate (and interesting), together with a proposed explanation as to why and how such measures could impact infant feeding practices. Likewise, the term “telehealth” and the extent and modality of its implementation should be described in greater detail.

RESPONSE: In the revised manuscript, we made an effort to provide more details on both the specific types of COVID-19 restrictions and recommendations for use of telehealth (including in pediatric patients) in Poland.

Moreover, “infant feeding practices” is a rather broad concept, especially when considering the age range of the infants included in the study, and should be further explained.

RESPONSE: In the revised manuscript, it was changed to “complementary feeding practices”. This change is consistent with the feeding practices for the age range of the infants (4-12 months of age) whose parents were surveyed. In line with the European (ESPGHAN) and similar Polish guidelines, complementary foods should not be introduced before 4 months but introduction should not be delayed beyond 6 months. At around 12 months of age, children are usually offered the same foods that the rest of the family eats.

Additionally, an adequate and relevant literature review, useful to “set the ground” and explain why the present study was “necessary”, is lacking.

RESPONSE: In the revised manuscript, we have added discussion of studies assessing the impact of the COVID-19 pandemic on breastfeeding. To our knowledge (and literature search), the impact of COVID-19 restrictions on complementary feeding practices has not previously been assessed. This makes the results of our study both novel and a necessary addition to the literature.

MATERIALS AND METHODS

The authors should include enough concise information in the methods section that it would be reproducible by other researchers.

Questionnaire design: A more detailed description of how the questionnaire was created would be of interest to the readers. What are the specific competences of the team who developed the questionnaire?

RESPONSE: Done. The questionnaire was developed by the study team, including two pediatric gastroenterologists with experience in infant nutrition research and a dietitian.

It would be interesting to know if and how the authors tested content validity. Furthermore, I wonder if the survey was pilot-tested before the beginning of the present study in order to assess understanding of the questions (if so, please describe).

RESPONSE: In the revised manuscript, the following sentence was added: “The validity of questionnaire was not tested, and a pilot survey was not performed.”

It is not clear why the questionnaire was proposed during two time periods. What was the difference between the two periods (if any)? What was the aim of said distinction if no comparison between the two “phases” was made? What restrictive measures were applied during the two time periods? Please explain.

RESPONSE: In the revised manuscript, in the Methods, we have added the rationale for having two data collections periods. Initially, we planned to compare the responses during the two collection periods. However, we abstained from such comparison for two reasons. First, unexpectedly, the response rate during the second collection period was much slower, even though the same strategy was used (as described in the Methods). Second, the COVID-19 restrictions differed throughout the country; however, the survey did not allow us to collect information on the type of restrictions.

It is not clear why the first period ended at 5000 responses, while the second one at way less (although remaining open for a longer period of time: 17 March to 13 May 2021 vs. 27 May to 3 June 2020). Please explain.

RESPONSE: Please see our response above and the revised manuscript for more details.

Outcomes: definitions of the outcomes are needed. What does the sentence “All outcomes were in the context of the COVID-19 restrictions” mean? Please explain.

RESPONSE: The outcomes in the revised manuscript were specified more accurately.

Participants: on which grounds was the age range (4-12 months) set? Please explain.

RESPONSE: Done. The aim of the study was to assess the impact of the COVID-19 restrictions on complementary infant feeding practices. In line with the European (ESPGHAN) and similar Polish guidelines, complementary foods should not be introduced before 4 months but introduction should not be delayed beyond 6 months. At around 12 months of age, children are usually offered the same foods that the rest of the family eats. In the revised manuscript, this information was clarified in the Methods.

Results: “Data from the online questionnaire were available from 6934 responders…who met the inclusion criteria”: it is not clear which exactly were the inclusion criteria. Were there any exclusion criteria? Please explain.

RESPONSE: As stated in the Methods (under Participants), the responders were ‘parents of infants aged 4 to 12 months (regardless of age, gender, race, ethnicity, level of education, financial situation, place of residence) and speaking Polish’. Only the inclusion criteria were prespecified.

Participants characteristics: what does “tertiary level of education” mean? Please explain. 

RESPONSE: Tertiary level of education means subjects with university diplomas. This information is clarified in the revised manuscript.

What is the significance of the question on the impact of COVID-19 on participants’ financial situation? How does it fit with the declared aim of the study?

RESPONSE: Financial status and its change due to COVID-19 may have an impact on infant feeding practices, i.e., choices of cheaper foods, consumption of more home-made foods. This information is included in the Discussion.

Survey responses: a bulleted list does not seem to be the best way to describe study results in a scientific publication. I would suggest you rewrite these results in a more discursive manner.

RESPONSE: All the bullets were removed.

Why were the three healthcare/telehealth-related questions added in the second time period only? What changed between the first and the second period? Was telehealth not implemented during the first time period? Please explain.

RESPONSE: The use of the telehealth was more common during the second survey interval. In the revised manuscript, this is explained in the Methods.

Tables: what do “vocational” and “answer denial” mean? Please find better suited synonyms.

RESPONSE: ‘Answer denial’ was changed to ‘none declared’. According to the Cambridge dictionary, ‘vocational’ is defined as “providing skills and education that prepare you for a job.” We do not include this definition in the manuscript but would be happy to add it if the Reviewer thinks this is necessary.

Discussion – Summary of main results: Reference to the data currently available in the literature is interesting, even if not always appropriate or well explained.

RESPONSE: While the nature of the Reviewer’s concern is unclear, we have revised this section to include our new results (multivariate logistic regression analysis). Based on a study in the current literature, we have also mentioned that the impact of COVID-19 restrictions, (i.e., loss of household income and/or limited access to fresh and safe foods) mostly affects the most vulnerable groups, including those of lower household income, Black and other ethnic minority groups, and/or parents with a low level of education, thus, further studies assessing the way the that child nutrition has changed in these groups most affected by COVID-19 restrictions are needed.

Sources of information: the literature review seems a bit off-topic with respects to the declared aim of the study and the conclusion that “Overall, both pandemic and pre-pandemic data document that there is a need to increase parents’ eagerness to look for professional support (i.e., pediatrician, dietitian, midwifery)” is beyond the scope of the present study. I would suggest a more focused literature review and comparison with the present study findings.

RESPONSE: As suggested by the Reviewer, we have modified the Discussion. Specifically, we focused on the COVID-19 pandemic data.

Impact of (--> on) infant feeding practices: since this is one of the main outcomes of the study, a more detailed literature review would seem appropriate. The only study referenced is a 2020 UK paper by Vazquez-Vazquez et al., which the present study seems largely inspired to.

RESPONSE: We did our best to provide a more detailed and focused literature review.  Unfortunately, there is no similar research on complementary feeding practices. Thus, comparison with other studies is limited. However, we tried to explain why, in the assessed population, a lack of impact of COVID-19 restrictions on infant feeding has been observed and reported.

Food supplements: the sentence “However, a recent large observational study, which included data from more than 400,000 COVID-19 Symptom Study app users, found that women taking multivitamins, omega-3 fatty acids, probiotics, or vitamin D were less likely to test positive for SARS-CoV-2. Randomized controlled trials to assess the efficacy of selected food supplements for preventing COVID-19 are needed” is completely off-topic. I would suggest you remove it.

RESPONSE: Done.

Conclusions: the conclusions should be rewritten. In their current form they do not offer a clear conclusion to the study, nor they reflect the study findings. Since the Authors found that “No major issues regarding the impact of the COVID-19 restrictions on the way of feeding, changes in feeding patterns, or complementary feeding introduction were reported”, it is not clear how they can conclude that “improving understanding of how the COVID-19 restrictions impact infant feeding practices could be of value when planning further quality pediatric healthcare”.

RESPONSE: The Conclusions section was rewritten.

Additional considerations: some minor mistakes are present, and the paper would benefit from being revised and possibly copyedited by a native English speaker, to improve its readability. In particular, the way dates are written should be revised.

RESPONSE: The manuscript was edited by a native English speaker with a medical background (M.D.). If there are any remaining issues, please let us know.

Reviewer 2 Report

This is an interesting manuscript, with a large, though not representative, sample that seeks to understand how lockdowns has impacted on infant (under 12 months) feeding.

The introduction provides a framework on restrictions in Poland and their possible relation to changes in infant feeding. However, I think it is necessary to introduce a few paragraphs to summarise what has already been published on restrictions and infant feeding from the experience of other countries (e.g. Brown and Shenker in the UK, year 2020, or Latorre et al in Italy, year 2021, as well as other studies mentioned in the discussion section such as the New Mum in the UK).

The methodology describes the research process undertaken. As it was an online survey, it is necessary to argue why there was no other way to collect the sample at that time (lockdown, etc...). In addition, it would be useful to provide more information on how the survey was disseminated, how participation was reinforced, why it was decided to repeat it with the second sample, why it is thought that 5000 were reached quickly in the first sample but only 1934 in the second, etc. Although some more information is then provided in the limitations section, it is still little information. 

The results are clearly shown in the results section. However, in my opinion, more statistical calculations are needed. First of all, you cannot add the 5000 and the 1934 without first testing whether the results are similar or different between the two samples, since they were collected at two different points in time.  It is like adding two cross-sectional studies. If the results are similar in each group, go ahead and add the results together. If they are different, present them separately and comment in the discussion on what might account for the changes.
Secondly, in the Brown and Senkher study, it is shown that mothers with low socio-economic status have been the most affected in relation to breastfeeding by the restrictions due to Covid. The results need to be compared according to the socio-economic status of the mothers. As the sample is so skewed towards mothers of high socio-economic status, the conclusions may not be correct for the segment of mothers with lower socio-economic status.
Finally, many variables have been collected by which the results could be segmented (age of children, residence in rural or urban areas, age of mothers, educational level, etc.). I do not understand why a multivariate logistic regression analysis, or at least a bivariate analysis segmenting the results by these socio-demographic variables, has not been carried out. I think that with such a large sample, it would be very easy and would significantly improve the quality of the results.

The discussion section is appropriate for the analyses undertaken. But I believe that with the new analyses it could be expanded much further. 
The conclusions could be improved by writing another paragraph beforehand with a summary of the main results after the new analyses.

Author Response

This is an interesting manuscript, with a large, though not representative, sample that seeks to understand how lockdowns has impacted on infant (under 12 months) feeding.

The introduction provides a framework on restrictions in Poland and their possible relation to changes in infant feeding. However, I think it is necessary to introduce a few paragraphs to summarise what has already been published on restrictions and infant feeding from the experience of other countries (e.g. Brown and Shenker in the UK, year 2020, or Latorre et al in Italy, year 2021, as well as other studies mentioned in the discussion section such as the New Mum in the UK).

RESPONSE: We appreciate the comment by the Reviewer. As suggested, the studies mentioned by the Reviewer were briefly summarized in the Introduction and in the Discussion.

The methodology describes the research process undertaken. As it was an online survey, it is necessary to argue why there was no other way to collect the sample at that time (lockdown, etc...).

RESPONSE: An explanation was added as the following sentence: “An online survey was chosen as the format due to the pandemic situation and the resulting COVID-19 restrictions.”

In addition, it would be useful to provide more information on how the survey was disseminated, how participation was reinforced, why it was decided to repeat it with the second sample, why it is thought that 5000 were reached quickly in the first sample but only 1934 in the second, etc. Although some more information is then provided in the limitations section, it is still little information. 

RESPONSE: In the revised manuscript, the reason for two data collection (sample) periods is explained. The reason for the lower recruitment rate during the second sample period is difficult to assess. However, it may be associated with the lack of willingness of parents to participate in this part of the study due to the probably larger number of online surveys during the pandemic.

The results are clearly shown in the results section. However, in my opinion, more statistical calculations are needed. First of all, you cannot add the 5000 and the 1934 without first testing whether the results are similar or different between the two samples, since they were collected at two different points in time. It is like adding two cross-sectional studies. If the results are similar in each group, go ahead and add the results together. If they are different, present them separately and comment in the discussion on what might account for the changes.

RESPONSE: We appreciate the comment made by the Reviewer. As explained in the revised manuscript, the success with the recruitment stimulated us to repeat the survey (with three additional questions related to telehealth reflecting changing healthcare environment). We planned to compare the responses during the two data collection periods, assuming it would allow us to analyze how the COVID-19 restrictions impact complementary feeding practices. During the second data collection period, the same number of responses was expected. However, unexpectedly, the recruitment was slower, and access to the questionnaire was closed before the target sample size was reached. Additionally, whereas the study periods coincided with the timing of the COVID-19 restrictions, neither of the study periods exactly coincided with the strict lockdown, which occasionally varied across the country. Considering the above, we abstained from comparing the two study periods.

Secondly, in the Brown and Senkher study, it is shown that mothers with low socio-economic status have been the most affected in relation to breastfeeding by the restrictions due to Covid. The results need to be compared according to the socio-economic status of the mothers. As the sample is so skewed towards mothers of high socio-economic status, the conclusions may not be correct for the segment of mothers with lower socio-economic status.

RESPONSE: We agree with the Reviewer on the importance of the socio-economic status. However, to perform such analyses, more detailed data characterizing the socio-economic status, such as income, education, and occupation, are needed. Unfortunately, we did not collect such data. Thus, we prefer to abstain from the analysis suggested by the Reviewer, as it may be misleading. Still, we thank the Reviewer for this important comment. In the revised manuscript, we referred to this Brown and Shenker study and their findings in the Introduction.

Finally, many variables have been collected by which the results could be segmented (age of children, residence in rural or urban areas, age of mothers, educational level, etc.). I do not understand why a multivariate logistic regression analysis, or at least a bivariate analysis segmenting the results by these socio-demographic variables, has not been carried out. I think that with such a large sample, it would be very easy and would significantly improve the quality of the results.

RESPONSE: As suggested by the Reviewer, multivariate logistic regression analysis was performed using the variables suggested.

The discussion section is appropriate for the analyses undertaken. But I believe that with the new analyses it could be expanded much further. 

RESPONSE: Done.

The conclusions could be improved by writing another paragraph beforehand with a summary of the main results after the new analyses.

RESPONSE: A summary of the findings, including the findings from multivariate logistic regression, is presented in the first paragraph of the Discussion. Thus, we abstained from repeating them in the Conclusions. However, we have revised the Conclusions based on comments by the other Reviewer.

Round 2

Reviewer 1 Report

The manuscript has been revised and improved. However, some minor revisions are still needed:

  1. I believe that writing “Complementary feeding practices” would suffice, and I would suggest you remove “infant” (i.e, “complementary infant feeding practices”).
  2. It is still not clear how the Authors can state in the conclusions that “the results of this online cross-sectional survey increase understanding of how COVID-19 restrictions in Poland impact complementary infant feeding practices”, since in the results they declare that “the COVID-19 restrictions had no impact on feeding practices in Poland, including the introduction of complementary feeding”. What I believe the study results actually clarified are the major issues associated with complementary feeding practices during the implementation of the COVID-19 restrictions in Poland.
  3. Although the Authors declared in their responses that they had the manuscript copy-edited by a native English speaker, the sentences added to the revised manuscript are not always grammatically nor syntactically correct. I would suggest you do some more editing of English language before publication.
  4. Please check lines 333-334: there seems to be some kind of error.

Author Response

1. Done.

2. We have changed the conclusion in the Abstract and the Conclusions, as suggested by the Reviewer.

3. The manuscript was once again edited by a native English speaker with a medical background (M.D.). If there are any remaining issues, please let us know.

4. Corrected.

Reviewer 2 Report

The changes made have improved the quality of the article. Congratulations on the work done.

I believe that a final effort is necessary to complete the results presented in the multivariate analysis.

Firstly, data have been collected for two variables related to the financial status of families, but have not been included in the multivariate analysis. I recommend choosing one of the two variables (financial family status or impact of covid19 pandemic on family material (financial) situation) and including it in all tables together with age, education and city of residence, and commenting on the results in the discussion, if appropriate.

Secondly, I recommend including another table, to include other variables where descriptive data are presented in table 2 but not included in the multivariate analysis. The missing variables are: Change in diet, Food supplements, Point of contact and Feeding related issues, and commenting on the findings in the discussion, if appropriate.

I believe that by doing these analyses the article would be complete and no further changes would be needed.

Author Response

I believe that a final effort is necessary to complete the results presented in the multivariate analysis.

Firstly, data have been collected for two variables related to the financial status of families, but have not been included in the multivariate analysis. I recommend choosing one of the two variables (financial family status or impact of covid19 pandemic on family material (financial) situation) and including it in all tables together with age, education and city of residence, and commenting on the results in the discussion, if appropriate.

Secondly, I recommend including another table, to include other variables where descriptive data are presented in table 2 but not included in the multivariate analysis. The missing variables are: Change in diet, Food supplements, Point of contact and Feeding related issues, and commenting on the findings in the discussion, if appropriate.

RESPONSE: As suggested by the Reviewer, multivariate analysis related to the family financial status, as well as the other identified missing variables, was done and findings were included (in the Results and the Discussion).

I believe that by doing these analyses the article would be complete and no further changes would be needed.

RESPONSE: Thanks.